# Protective Role of Hydrogen Gas on Oxidative Damage and Apoptosis in Intestinal Porcine Epithelial Cells (IPEC-J2) Induced by Deoxynivalenol: A Preliminary Study

**DOI:** 10.3390/toxins12010005

**Published:** 2019-12-19

**Authors:** Xu Ji, Weijiang Zheng, Wen Yao

**Affiliations:** 1Laboratory of Gastrointestinal Microbiology, Jiangsu Key Laboratory of Gastrointestinal Nutrition and Animal Health, College of Animal Science and Technology, Nanjing Agricultural University, Nanjing 210095, China; 2015205021@njau.edu.cn (X.J.); zhengweijiang@njau.edu.cn (W.Z.); 2National Experimental Teaching Center for Animal Science, College of Animal Science and Technology, Nanjing Agricultural University, Nanjing 210095, China; 3Key Lab of Animal Physiology and Biochemistry, Ministry of Agriculture, Nanjing 210095, China

**Keywords:** hydrogen gas, deoxynivalenol, IPEC-J2, oxidative damage, apoptosis

## Abstract

To explore the protective role of hydrogen gas (H_2_) on oxidative damage and apoptosis in intestinal porcine epithelial cells (IPEC-J2) induced by deoxynivalenol (DON), cells were assigned to four treatment groups, including control, 5 μM DON, H_2_-saturated medium, and 5 μM DON + H_2_-saturated medium treatments. After 12 h of different treatments, the cell viability, biomarkers of cell redox states, and gene expression of antioxidant enzymes and apoptosis were observed and detected. Furthermore, caspase-3 and Bax protein expressions were measured by Western blot analysis. Our results demonstrated that the 5 μM DON significantly caused cytotoxicity to IPEC-J2 cells by reducing cell viability and increasing lactate dehydrogenase release in culture supernatants. Moreover, DON treatments significantly increased levels of 8-hydroxy-2′-deoxyguanosine, 3-nitrotyrosine, and malonaldehyde; however, they decreased total superoxide dismutase and catalase activities and downregulated messenger RNA (mRNA) expression related to antioxidant enzymes in cells. The 5 μM DON treatment also downregulated Bcl-2 expression and upregulated caspase-3 and Bax expression. However, the H_2_-saturated medium significantly improved cell growth status and reversed the change of redox states and expression of genes and proteins related to apoptosis induced by DON in IPEC-J2 cells. In conclusion, H_2_ could protect IPEC-J2 cells from DON-induced oxidative damage and apoptosis in vitro.

## 1. Introduction

Mycotoxins are toxic secondary metabolites from fungi which often exist as contaminants in animal and human food worldwide. The consumption of mycotoxin-contaminated feed and food is considered a major health risk to animals and humans [1]. Deoxynivalenol (DON) belongs to the trichothecene family, and it is mainly generated from *Fusarium graminearum* or *Fusarium culmorum* [2], which are easily detected in some agricultural commodities, such as barley, wheat, or oat [3]. Ma et al. investigated the contamination of DON in foodstuffs from different provinces in China between 2016 and 2017. They found that the occurrence rate of DON was over 74.5%, in which the average concentration ranged from 450.0–4381.5 μg/kg, suggesting that DON was a prevalent contaminant in China [4].

Owing to the prevailing existence of DON in cereal grains, pigs are the most sensitive species when exposed to this mycotoxin. Following ingestion of a DON-contaminated diet, a reduction in growth and immunomodulating properties is induced [5]. The gastrointestinal tract is the primary target organ, and it is often exposed to high levels of toxic substances, where DON is rapidly absorbed by the epithelial surface [6]. In intestinal epithelial cells, DON can induce inflammation and oxidative stress, thereby accelerating cell apoptosis and influencing intestinal epithelial cell growth and function [7,8,9]. Therefore, providing a novel solution to improve mycotoxin-induced toxic effects on the intestine is growing more and more essential. 

Molecular hydrogen (hydrogen gas or H_2_) was historically considered as an inert and non-functional gas [10]. However, a notable capacity that hydrogen can distinctively neutralize •OH and ONOO^−^ was confirmed in 2007 [11]. Since then, further studies revealed its crucial biological roles in various types of disease models, including anti-oxidant, anti-apoptotic, and anti-inflammatory effects [12,13]. In particular, it had the capacity to attenuate some serious intestinal diseases [14,15,16]. There are efficient approaches to provide hydrogen in vivo when used for therapeutic effects, such as the inhalation of 1–4% hydrogen gas, drinking of hydrogen-rich water, injection of hydrogen-saline, and diffusion through the skin [11,13]. In addition, some studies showed that hydrogen directly displayed biological effects in cells in vitro. For example, Li et al. reported that H_2_-saturated medium ameliorated high glucose-induced oxidative stress and apoptosis in Schwann cells by inhibiting the production of •OH and ONOO^−^, caspase-3 activity, and apoptosis in Schwann cells [17]. H_2_-saturated medium also ameliorated oxidative stress in human skin fibroblasts caused by high glucose or mannitol [18]. It was shown that molecular hydrogen significantly decreased the intracellular O_2_^−^ level, as well as the production of 8-hydroxy-2′-deoxyguanosine (8-OHdG), 3-nitrotyrosine (3-NT), and malonaldehyde (MDA). In addition, the antioxidant system was improved with H_2_-saturated medium by increasing the activity of superoxide dismutase (SOD) and glutathione (GSH) [18]. Xie et al. also found that H_2_ neutralized •OH free radicals by enriching protein expression in the Nrf2/HO-1 signaling pathway in glucose deprivation-stimulated H9c2 cardiomyoblasts [19].

Intestinal porcine epithelial cells (IPEC-J2) are isolated from a non-transformed porcine intestinal columnar epithelial cell line derived from a neonatal piglet mid-jejunum, and they display similar properties to the intestinal epithelium [20]. Recent studies verified the toxic effects of DON on porcine intestinal epithelial cells when used with IPEC-J2 cells [7,21,22]. We previously reported that the oral administration of hydrogen-saturated water can moderately compensate grow suppression and intestinal damages in piglets induced by a *Fusarium* mycotoxin-contaminated diet [23,24]. Therefore, IPEC-J2 cells are very suitable for exploring whether hydrogen may directly have protective effects against oxidative damage and apoptosis induced by DON in vitro. Furthermore, this study may provide some valuable insights into hydrogen as a protective agent to ameliorate intestinal damage induced by mycotoxins in swine production.

## 2. Results

### 2.1. The Effects of DON on the Growth of IPEC-J2 Cells

To observe the cytotoxic effects of DON on the growth of IPEC-J2 cells, we firstly evaluated cell viability using the Methyl Thiazolyl Tetrazolium (MTT) assay. The results showed that DON at 5 μM, 10 μM, or 30 μM induced a dramatic decrease in the IPEC-J2 cell viability compared to the control group at 12 h (*p* < 0.05). However, no difference was found among 5 μM, 10 μM, and 30 μM DON treatments (Figure 1). Accordingly, the DON concentration at 5 μM was used for subsequent experiments.

### 2.2. Effects of H_2_ on Cell Viability and Apoptosis in DON-Induced IPEC-J2 Cells

The protective effect of H_2_ on cell viability was measured by the MTT assay. Lactate dehydrogenase (LDH) activity in the culture medium can indirectly indicate cell death/apoptosis [25,26]. As shown in Figure 2A and 2B, in the DON group, 5 μM DON significantly weakened cell viability (*p* < 0.05) and increased the release of LDH (*p* < 0.05) compared to the control group. However, H_2_ efficiently reduced the LDH release (*p* < 0.05) and increased cell viability (*p* < 0.05) in the DON + H_2_ group, suggesting that H_2_ significantly alleviated DON-induced IPEC-J2 cell growth inhibition.

### 2.3. Effects of H_2_ on Oxidant and Antioxidant Status in DON-Induced IPEC-J2 Cells

The results in Figure 3 indicate that 5 μM DON significantly increased the concentrations of 8-OHdG, 3-NT, and MDA and decreased total superoxide dismutase (T-SOD) and catalase (CAT) activities in the DON group compared to the control group (*p* < 0.05). However, H_2_ could significantly decrease the levels of 8-OHdG, 3-NT and MDA, and enhance T-SOD and CAT activities in DON-induced cells (*p* < 0.05).

### 2.4. Effects of H_2_ on the Expression of Antioxidant and Apoptosis Genes in IPEC-J2 Cells Exposed to DON

As shown in Figure 4A, compared to the control group, CAT, Mn-SOD, and CuZn-SOD gene expression levels were significantly downregulated in 5 μM DON-treated IPEC-J2 cells (*p* < 0.05). Compared with the DON group, incubating IPEC-J2 cells with H_2_-saturated medium significantly reversed the decreased expression of CAT and Mn-SOD for the effect of DON treatment in the DON + H_2_ group (*p* < 0.05). The messenger RNA (mRNA) expression of CuZn-SOD was also increased in the DON + H_2_ group, whereas no difference was found between the DON group and DON + H_2_ groups (*p* > 0.05). The results of apoptosis gene expression are shown in Figure 4B. The DON treatment significantly increased caspase-3 and Bax gene expression (*p* < 0.05) and decreased Bcl-2 gene expression compared to the control group (*p* < 0.05). Nevertheless, H_2_ significantly reduced caspase-3 and Bax expression (*p* < 0.05) and increased Bcl-2 gene mRNA expression (*p* < 0.05) in the DON + H_2_ group.

### 2.5. H_2_ Reduces DON-Induced Increase of Pro-Apoptosis Caspase-3 and Bax Protein Expression in IPEC-J2 Cells

The effects of H_2_ on caspase-3 and Bax protein expression in IPEC-J2 cells were detected by the Western blot technique. The results in Figure 5A show that caspase-3 protein expression level in the DON group was remarkably higher than in the control group (*p* < 0.05). However, the DON + H_2_ group had a lower caspase-3 protein level than the DON group (*p* < 0.05). Bax protein expression is shown in Figure 5B. We did not observe any significant difference among control, H_2_, and DON + H_2_ groups, whereas the DON + H_2_ group had a higher Bax protein expression level than the other three treatments (*p* < 0.05).

## 3. Discussion

The intestinal epithelium is a single-cell layer which has two main functions: absorbing useful substances into the body and providing a barrier against the entry of harmful substances in the intestinal lumen, such as some pathogens and food contaminants [27]. After the pigs ingest a DON-contaminated diet, the intestinal epithelium is the specific site where DON is principally absorbed. It was shown that DON can destroy the protein structure of intestinal epithelial cells and the intestinal barrier integrity by binding to ribosomes, resulting in “ribosomal stress” and inhibiting protein synthesis [28,29,30]. Some studies indicated that DON mainly induced damage to the jejunum structure of pigs and caused oxidative stress and apoptosis in jejunum epithelial cells [31]. IPEC-J2 is commonly utilized to analyze the transport function of intestinal epithelium and the influence of nutrients or toxins on intestinal function [20]. Accordingly, IPEC-J2 is a good model to explore cytotoxic properties of mycotoxins on intestinal epithelial cells in vitro. Recent reports showed that hydrogen can remit some serious intestinal diseases with different methods of intake into the body [14,15,16]. In addition, its protective effects were demonstrated in some cell models in vitro with the use of H_2_-saturated medium [32,33,34]. Therefore, we used DON, which is a prevalent mycotoxin in swine production, to induce cell injury to investigate the protective effects of H_2_ on oxidative damage and apoptosis in IPEC-J2 cells.

In the present research, we firstly accessed the cytotoxic effects of DON in IPEC-J2 cells. The results showed that 5 μM DON (1481.6 ng/mL) could cause a dramatic decrease in cell viability when incubated with IPEC-J2 for 12 h. Previous studies indicated that the DON concentration was approximately 0.44 μM (0.13 mg/kg) in the fresh chyme of the small intestine when pigs (88 kg body weight) received 1.1 kg of feed with 4.2 mg/kg DON [35]. However, the selected DON concentration used in cell culture experiments is difficult to correlate with in vivo exposure. This is because purified mycotoxins are usually used in cell research, whereas naturally contaminated feed ingredients are often used in vivo experiments. Moreover, the absorption amount of mycotoxin by cells in vitro also hardly corresponds to the amount absorbed from feed in vivo. Some reports suggested that 10–30 μM DON used in vitro can correspond to 3–10 mg/kg of food or feed [36]. A survey of DON concentration in complete pig feed from 2016 to 2017 in China suggested that the mean values were around 753.1–1194.0 μg/kg [4]. Therefore, the DON concentration of 5 μM (1481.6 ng/mL) with IPEC-J2 cells in vitro can partly model the feeding situation when pigs are generally given a DON-contaminated diet in China. Based on the above results, 5 μM DON-treated cells for 12 h were used to establish the cellular injury model in the subsequent experiments.

It was shown that trichothecene mycotoxins can induce cellular oxidative stress, such as accumulation of ROS, increasing lipid peroxidation, and affecting the antioxidant system [37]. Among ROS, •OH and ONOO^−^ are cytotoxic to cells and rapidly induce cell damage; however, they are exceedingly unsteady in living cells because of a short biological half-life caused by their strong oxidative capabilities [38]. Therefore, we measured the levels of 8-OHdG and 3-NT which are oxidative products of •OH and ONOO^−^ [39,40]. The results showed that 8-OHdG and 3-NT concentrations were increased in DON-treated IPEC-J2 cells. These findings were in accordance with an earlier study that found that exposure of human hepatoma HepG2 cells to 7.5 μM DON caused a significant increase in cellular 8-OHdG [2]. Kang et al. also reported that both 1000 ng/mL and 2000 ng/mL DON promoted excessive ROS production in IPEC-J2 cells [41]. Furthermore, the antioxidant capacity may be weakened when cellular oxidative stress is significantly enhanced in cells. Our results showed that T-SOD and CAT activities were all reduced in DON-induced IPEC-J2 cells. The cellular concentration of MDA was also increased when treated with DON. MDA originates from lipid peroxidation, which suggests that cellular oxidative defense is inadequate [18]. Previous studies illustrated that 1000 ng/mL DON significantly reduced T-SOD and CAT activities in IPEC-J2 cells [41]. Kuoadio et al. also found 10 μM DON increased the MDA level in Caco-2 cells [42]. In addition, in our experiment, the gene mRNA expression of antioxidant enzymes (CAT, Mn-SOD, and CuZn-SOD) was also remarkably reduced in the presence of DON.

The principal molecular mechanism identified for hydrogen was its antioxygenation [43]. It was demonstrated that H_2_ can selectively neutralize •OH and ONOO^−^ in cells [11]. Unlike some antioxidants which cannot focus on organelles effectively, hydrogen can readily penetrate cell membranes and rapidly translocate into the cytosol, mitochondria, and nucleus to scavenge oxygen free radicals because of its simple molecular structure [44]. Our results also showed that H_2_-saturated medium could reduce the DON-induced increase of 8-OHdG and 3-NT, suggesting that the •OH and ONOO^−^ were neutralized in cells. •OH is also known as a major trigger of chain reactions of free radicals, such as lipid peroxidation. Thus, H_2_ has the advantage of inhibiting lipid peroxidation after neutralizing •OH in cells, which induces the production of oxidative stress markers (e.g., 4-hydroxyl-2-nonenal and MDA) [45,46,47]. Recent reports indeed indicated that the MDA content was decreased with molecular hydrogen in cholangiocyte cells induced by hypoxia/reoxygenation injury [48]. The current results also showed that H_2_ reduced the level of MDA in IPEC-J2 cells exposed to DON. Additionally, some animal experiments showed that H_2_ can improve the activity of antioxidant system, resulting in a decrease in damage to cells and tissues caused by oxidative stress [49]. Yu et al. reported that H_2_-saturated medium increased SOD and GSH activities, enhancing the antioxidative defense system in human skin fibroblasts exposed to high glucose or mannitol [18]. In our study, similar results were found, whereby the treatment with H_2_ increased the expression levels of CAT and Mn-SOD genes with CAT and T-SOD activities in IPEC-J2 challenged by DON. We speculated that the activities of antioxidative enzymes were protected after H_2_ neutralized •OH and ONOO^−^ in IPEC-J2 cells. Therefore, the potential motivating mechanisms need to be explored in the future.

Trichothecenes generate free radicals, reduce antioxidant enzymes activities, and ultimately induce apoptosis in cells [37]. A previous report indicated that 10 μM DON had toxic effects on IPEC-J2 cells, which caused cell rounding, autolysis, and detachment from the monolayer, also increasing LDH release into the medium [26]. In the present study, we also observed cell detachment from the epithelial monolayer stimulated by DON, which was accompanied by a decrease in cell viability and an increase in LDH activity in culture supernatants. This suggested that DON significantly caused damage and apoptosis in IPEC-J2 cells. The activation of caspase-3 and Bcl-2 family protein expression is related to apoptosis induced by DON [50]. In previous studies, the expression of caspase-3 was significantly increased after treatment with 500 ng/mL DON [41]. Also, 20 μM DON significantly affected rat (IEC-6) intestinal epithelial cell viability through a pro-apoptosis process which involved the inhibition of anti-apoptosis protein Bcl-2 and induction of pro-apoptosis protein Bax [51]. Our results also showed that DON treatment downregulated the mRNA expression of Bcl-2; however, it upregulated the mRNA and protein expression of caspase-3 and Bax.

According to recent studies, the anti-apoptosis effect of H_2_ was confirmed in some animal and cell models. For example, our previous study demonstrated that oral administration of hydrogen-saturated water reduced apoptosis and gene expression related to the apoptosis of epithelium cells in the jejunum and ileum of piglets fed mycotoxin-contaminated diets [23]. Moreover, Jiang et al. suggested that H_2_-saturated medium suppressed advanced glycation end product-induced apoptosis and upraised the Bcl-2/Bax ratio in endothelial cells [52]. It was also shown in the present study that H_2_ protected IPEC-J2 cells against a decline in cell viability and an increase in LDH release in medium exposed to DON. Furthermore, the caspase-3 and Bax gene and protein expressions were downregulated, and the mRNA expression level of Bcl-2 was upregulated by the use of H_2_-saturated medium. However, the exact mechanisms underlying the anti-apoptosis effect of H_2_ are still unclear. It may be due to molecular hydrogen having the capability to neutralize toxic free radicals, protect DNA and protein, and then inhibit apoptosis in cells. Further studies are warranted to explore some special pathways in this process.

## 4. Conclusions

The present results indicate that hydrogen decreased the levels of •OH, ONOO^−^, and MDA, as well as increased the gene expression and activities of antioxidative enzymes in IPEC-J2 cells stimulated by DON. In addition, hydrogen ameliorated DON-induced apoptosis through the increase in Bcl-2 expression and the decrease in Caspase-3 and Bax expression. Collectively, this study suggests that H_2_ application precisely protects IPEC-J2 cells against oxidative damage and apoptosis induced by DON in vitro. Furthermore, the present in vitro research and previous results in piglets [23,24,53] could potentially contribute to the implication and application of H_2_ as a novel solution to alleviate intestinal damage induced by mycotoxins in swine production in the future.

## 5. Materials and Methods

### 5.1. Cell Culture

The IPEC-J2 cell line was kindly supplied by Prof. Qian Yang from Nanjing Agricultural University (Nanjing, China) [54]. The cells were normally cultured in Dulbecco’s modified Eagle medium/nutrient mixture F-12 (DMEM/F-12; Invitrogen, Shanghai, China), which contained 10% fetal bovine serum (FBS; Cellgro, Mediatech, Herndon, VA, USA), 100 U/mL penicillin, and 100 μg/mL streptomycin (Invitrogen, Shanghai, China) at 37 °C in 5% CO_2_. The cell culture medium was changed every two days. When the cell confluence reached approximately 80%, 0.25% trypsin-EDTA (Invitrogen, Shanghai, China) was used to digest and passage cells. We selected passages 2–5 for subsequent experiments.

### 5.2. Preparation of H_2_-Saturated Medium

The preparation and storage of H_2_-saturated medium were according to previous studies with some modifications [33]. Hydrogen was obtained from a hydrogen gas tank (≥99.999% purity) and then it was dissolved in serum-free DMEM/F12 in a Pytex bottle for 30 min under 0.2 MPa pressure to get a supersaturated H_2_ concentration (≥0.6 mM). In this process, we used a special catheter with a 0.2-μm filter (Acrodisc Syringe Filters with Supor Membrane, Pall Corporation, Ann Arbor, MI, USA) to remove bacteria. Subsequently, 10% FBS was added into the H_2_-saturated medium. Sealed aluminum bags were used for the storage of H_2_-saturated medium. The concentration of H_2_ in medium was determined by a hydrogen sensor (Unisense, Aarhus, Denmark) based on the method in our previous experiments [24]. When the H_2_-saturated medium was prepared freshly, its concentration was at 0.61 ± 0.03 mM. After 12 h of incubation, H_2_ concentration was measured again at 0.06 ± 0.00 mM. The same method was also used for preparing the nitrogen-saturated medium.

### 5.3. Preparation of DON

Purified DON (D0156; Sigma-Aldrich, Shanghai, China) was dissolved in DMSO (Sigma-Aldrich, Shanghai, China) to 30 mM (8889.6 μg/mL). The stock solutions were stored at −20 °C before dilution in cell culture medium. Working solutions of 30 μM (8889.6 ng/mL), 10 μM (2963.2 ng/mL), and 5 μM (1481.6 ng/mL) were prepared in the cell culture medium for the subsequent experiments. The DMSO was added as a control group. A final concentration of 0.1% DMSO corresponding to the highest DMSO concentration of working solutions was detected in all assays, and the results suggested that no significant difference was found from the control.

### 5.4. Estimation of Cytotoxic Effects of DON on IPEC-J2 Cells

Cell viability was assayed using an MTT cell proliferation and cytotoxicity assay kit (Sangon Biotech, Shanghai, China). IPEC-J2 cells were spilled into 96-well plates at 2 × 10^3^ cells/well in 100 μL of medium, and each group had six parallel wells. After the cells were cultured with 5 μM, 10 μM, or 30 μM DON for 12 h, 10 μL of MTT (5 mg/mL) solution was added to each well and continually incubated for 4 h. The culture medium was aspirated, and then 100 μL of formazan solubilization solution (Sangon Biotech, Shanghai, China) was added to each well. After the solution was mixed gently for 10 min on a shaker, the absorbance was determined at 570 nm by a spectrophotometer (Thermo Scientific, Vantaa, Finland).

### 5.5. Cell Treatments

To determine the protective effects of the H_2_-saturated medium, IPEC-J2 cells were divided into four groups: control group, 5 μM DON treatment (DON group), H_2_-saturated medium treatment (H_2_ group), and 5 μM DON and H_2_-saturated medium treatment (DON + H_2_ group). Briefly, when the IPEC-J2 cell confluence reached approximately 80%, in the control group, IPEC-J2 cells were incubated in nitrogen-saturated medium and DMSO was added. For the DON group, IPEC-J2 cells were kept in nitrogen-saturated medium and 5 μM DON was added. For the H_2_ group, IPEC-J2 cells were cultured in H_2_-saturated medium and DMSO was added. In the DON + H_2_ group, IPEC-J2 cells were incubated in H_2_-saturated medium and 5 μM DON was added. All cells in the four groups were cultured for another 12 h before subsequent experiments.

### 5.6. Cell Viability Assay

IPEC-J2 cells were spilled into 96-well plates at 2 × 10^3^ cells/well in 100 μL of medium, and each group had six parallel wells. After the cells were cultured with four treatments for 12 h as described above, 10 μL of MTT (5 mg/mL) solution was added to each well and continually incubated for 4 h. The culture medium was aspirated, and then 100 μL of formazan solubilization solution was added to each well. After the solution was mixed gently for 10 min on a shaker, the absorbance was determined at 570 nm by a spectrophotometer (Thermo Scientific, Vantaa, Finland).

### 5.7. LDH Activity Measurement

Cell culture supernatant was collected to assess lactate dehydrogenase (LDH) release due to cell death/apoptosis [25,26]. Briefly, IPEC-J2 cells (2 × 10^5^ cells/well) were grown in six-well plates. After confluence reached approximately 80%, the cells were cultured with the various treatments for 12 h. Subsequently, the culture supernatants in the four groups were collected. The LDH activity was detected with an LDH cytotoxicity assay kit (Nanjing Jiancheng Bioengineering Institute, Nanjing, China) at an optical density of 450 nm using a spectrophotometer (Thermo Scientific, Vantaa, Finland) following the manufacturer’s instructions.

### 5.8. 8-OHdG and 3-NT Concentration Detection

It is known that 8-OHdG and 3-NT are biomarkers of cell oxidant status, as they are the oxidative products of •OH and ONOO^−^ [38]. IPEC-J2 cells were incubated in six-well plates at 2 × 10^5^ cells/well. When confluence reached approximately 80%, different treatments were added to the cells for 12 h. Ultrasound technique was used for cell lysis. The supernatants were used to detect 8-OHdG and 3-NT concentration following centrifugation. Commercial ELISA kits (Nanjing Jiancheng Bioengineering Institute, Nanjing, China) were used referring to the manufacturer’s instructions.

### 5.9. Determination of Cellular MDA Level, and T-SOD and CAT Activities

IPEC-J2 (2 × 10^5^ cells/well) were grown in six-well plates. After confluence reached approximately 80%, they were cultured with four treatments for 12 h. Subsequently, the treated cells were collected and lysed by ultrasound. After centrifugation, MDA levels, and T-SOD and CAT activities were measured by assay kits (Nanjing Jiancheng Bioengineering Institute, Nanjing, China) referring to the manufacturer’s instructions.

### 5.10. Total RNA Isolation and Quantitative Real-Time PCR

As previously described, the IPEC-J2 cells were cultured in six-well plates (2 × 10^5^ cells/well), grown, and treated in four groups for 12 h. Total RNA was isolated cells by RNAiso Plus following the manufacturer’s instructions (Takara, Dalian, China). The concentration and purity of total RNA samples were determined by spectrophotometer (NANODROP 2000, Thermo Fisher Scientific, Waltham, MA, USA). For reverse transcription, a Prime ScriptTM RT reagent Kit with genomic DNA (gDNA) Eraser (Perfect Real Time) (Takara, Dalian, China) was used for generating complementary DNA (cDNA). SYBR Premix Ex TaqTM (Takara, Dalian, China) was used in quantitative real-time PCR following the manufacturer’s instructions. Gene specific primers for CAT, MnSOD, CuZnSOD, caspase-3, Bcl-2, and Bax (Table 1) were manufactured commercially (Invitrogen, Shanghai, China). The real-time PCR reactions were carried out in a QuantStudio 5 Real-Time PCR System (Thermo Fisher Scientific, Waltham, MA, USA) with the following cycle system: a recycling stage at 95 °C for 30 s, 40 cycles of denaturation at 95 °C for 5 s, and annealing at 60 °C for 30 s. For data normalization and analysis, beta-actin (β-actin) was used as the reference gene, and the relative target gene mRNA expression was calculated by the 2^−ΔΔCT^ method.

### 5.11. Western Blot Analysis

IPEC-J2 cells were also cultured with different treatments as previously described in six-well plates. All samples were lysed using cell lysis buffer (Beyotime, Nantong, China) based on the manufacturer’s instructions. Total protein concentration in samples was measured by a Bicinchoninic acid (BCA) protein assay kit (Thermo Fisher Scientific, Waltham, MA, USA). Twenty micrograms of protein was separated by electrophoresis in 12% Tris-glycine SDS running buffer and then transferred to Immun-Blot^®^ PVDF membranes (Bio-Rad, Hercules, CA, USA). Subsequently, the membrane was blocked with 5% nonfat dry milk for 1 h at room temperature and then incubated with caspase-3 (1:1000), Bax (1:1000), and β-actin (1:5000) at 4 °C overnight. After washing in TBST buffer, the membrane was incubated for 1 h at room temperature with secondary peroxidase-conjugated anti-rabbit or anti-mouse antibodies. Immuno-positive bands were visualized by the Versa DocTM imaging system after incubating in TanonTM High-sig ECL Western blotting substrate (Tanon, Shanghai, China) following the manufacturer’s instructions. The bands were analyzed by the ImageJ software (National Institutes of Health, Bethesda, MD, USA). Target protein expressions were normalized to β-actin and shown as the fold change compared to the control group.

### 5.12. Data Presentation and Statistical Analysis

The data in this study were expressed as means ± standard error of mean (SEM) of three independent experiments. SPSS software (Version 23.0, SPSS Inc., Chicago, IL, USA) was used to carry out all statistical analysis. One-way ANOVA was used to determine the differences among four groups followed by the Turkey–Kramer test for multiple comparisons. A *p*-value <0.05 was judged to show a statistically significant difference.

## Figures and Tables

**Figure 1 toxins-12-00005-f001:**
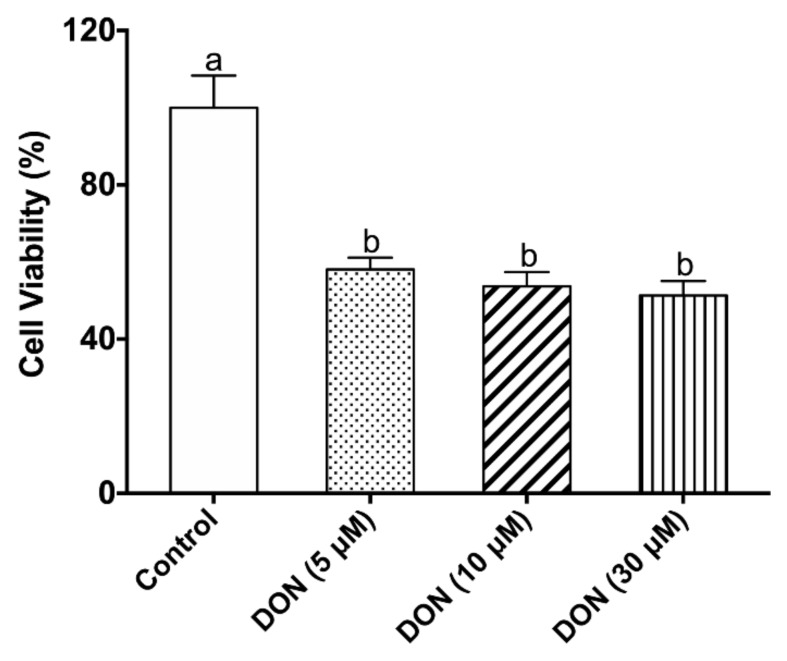
Effects of 5 μM, 10 μM, and 30 μM deoxynivalenol (DON) on the cell viability of intestinal porcine epithelial cells (IPEC-J2) cells for 12 h. The values stand for means ± standard error of the mean (SEM) of three independent experiments. Different letters on top of the bars denote a significant difference within the four groups (*p* < 0.05).

**Figure 2 toxins-12-00005-f002:**
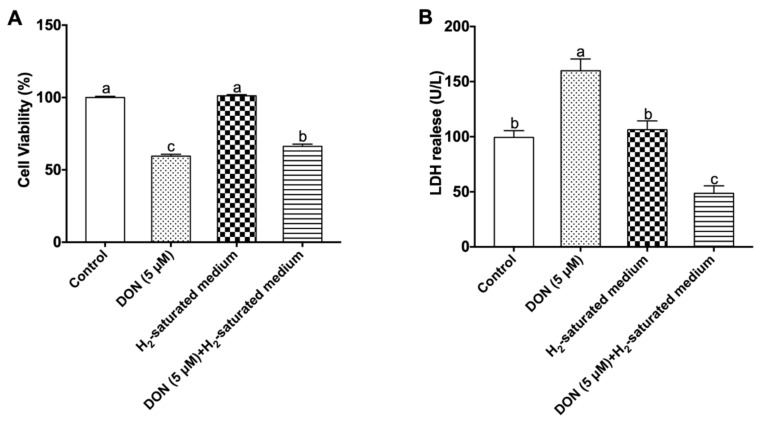
Effects of H_2_ on the cell viability and apoptosis in DON-induced IPEC-J2 cells: (**A**) Effects of H_2_ on IPEC-J2 cell viability challenged by DON; (**B**) effects of H_2_ on lactate dehydrogenase (LDH) release in the cell culture medium. The values stand for means ± SEM of three independent experiments. Different letters on top of the bars denote a significant difference within the four groups (*p* < 0.05).

**Figure 3 toxins-12-00005-f003:**
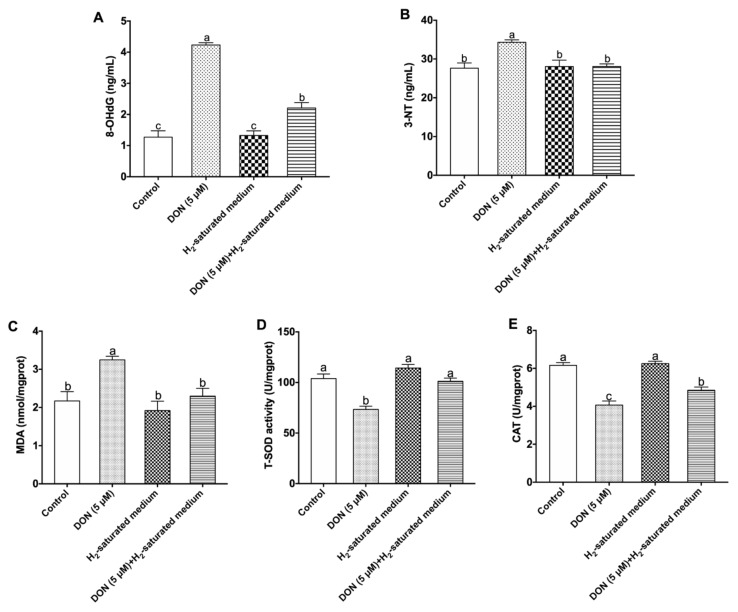
Effects of H_2_ on oxidant and antioxidant status in IPEC-J2 cells stimulated by DON: (**A**) 8-hydroxy-2′-deoxyguanosine (8-OHdG) concentration; (**B**) 3-nitrotyrosine (3-NT) concentration; (**C**) malonaldehyde (MDA) level; (**D**) total superoxide dismutase (T-SOD) activity; (**E**) catalase (CAT) activity in IPEC-J2 cells. The values stand for mean ± SEM of three independent experiments. Different letters on top of the bars denote a significant difference within the four groups (*p* < 0.05).

**Figure 4 toxins-12-00005-f004:**
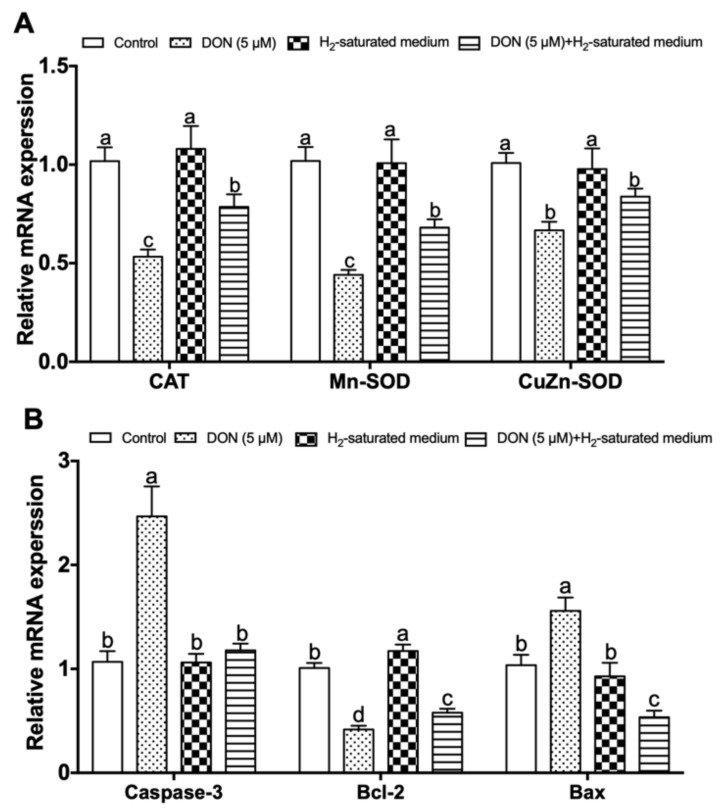
Effects of H_2_ on the expressions of antioxidant and apoptosis genes in DON-induced IPEC-J2 cells: (**A**) CAT, Mn-SOD, and CuZn-SOD gene expression related to antioxidant enzymes; (**B**) caspase-3, Bcl-2, and Bax gene expression related to apoptosis. The values stand for mean ± SEM of three independent experiments. Different letters on top of the bars denote a significant difference within the four groups (*p* < 0.05).

**Figure 5 toxins-12-00005-f005:**
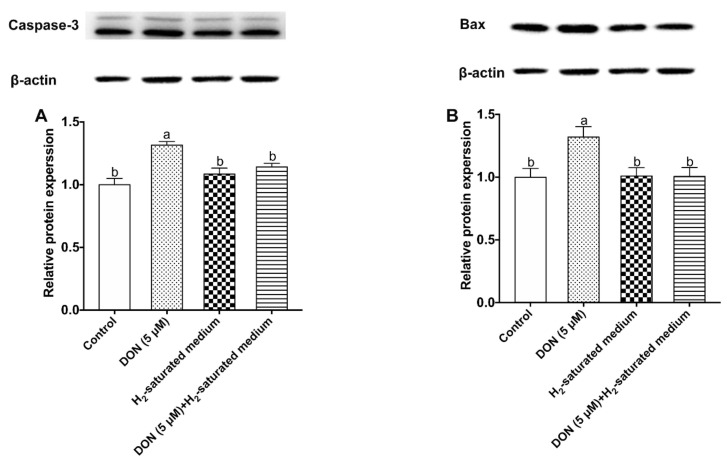
Effects of H_2_ on caspase-3 and Bax protein expression in IPEC-J2 cells challenged by DON: (**A**) caspase-3 protein expression; (**B**) Bax protein expression in IEPC-J2 cells. The values stand for means ± SEM of three independent experiments. Different letters on top of the bars denote a significant difference within the four groups (*p* < 0.05).

**Table 1 toxins-12-00005-t001:** List of RT-qPCR primer sequences used in this study. CAT—catalase; SOD—superoxide dismutase.

Target Genes	Primer Sequence (5′–3′)	Accession Number	Size
CAT	CTTGGAACATTGTACCCGCT	NM_214301.2	241
	GTCCAGAAGAGCCTGAATGC		
Mn-SOD	GGACAAATCTGAGCCCTAACG	NM_214127.2	159
	CCTTGTTGAAACCGAGCC		
CuZn-SOD	CAGGTCCTCACTTCAATCC	NM_001190422	255
	CCAAACGACTTCCASCAT		
Caspase-3	GTGGGACTGAAGATGACA	NM_214131.1	190
	ACCCGAGTAAGAATGTG		
Bcl-2	TCCAGAACCTCCTTGGTCCT	XM_021099593.1	187
	AACTACAGCGAGGTGCTTCC		
Bax	TTTCTGACGGCAACTTCAACTG	XM_003127290.5	236
	AGCCACAAAGATGGTCACTGTCT		
β-actin	GGACTTCGAGCAGGAGATGG	XM_003357928.4	233
	GCACCGTGTTGGCGTAGAGG		

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
