# Peer review of "Protective Role of Hydrogen Gas on Oxidative Damage and Apoptosis in Intestinal Porcine Epithelial Cells (IPEC-J2) Induced by Deoxynivalenol: A Preliminary Study"

_toxins, 2019, doi:10.3390/toxins12010005_

Round 1
Reviewer 1 Report
In recent years, the occurrence of mycotoxins in feed used for breeding livestock has been increasingly observed. Therefore, attempting to reduce oxidative damage and apoptosis of the porcine intestinal epithelium seems justified.
The work was well planned and described in a manuscript that is very easy to read and follow. In my opinion, it is recommended for publication after some minor corrections:
A few punctuation problems are present in the manuscript. I suggest the Authors to double-check the text.
Author Response
Response to Reviewer 1 Comments
Point 1: In recent years, the occurrence of mycotoxins in feed used for breeding livestock has been increasingly observed. Therefore, attempting to reduce oxidative damage and apoptosis of the porcine intestinal epithelium seems justified.
The work was well planned and described in a manuscript that is very easy to read and follow. In my opinion, it is recommended for publication after some minor corrections:
A few punctuation problems are present in the manuscript. I suggest the Authors to double-check the text.
Response 1: Dear Reviewer, thank you for your kindly comments and advice. We have double-checked the punctuation problems in our manuscript and modified them.
Reviewer 2 Report
General comment
The current manuscript addresses in an in vitro model with IPEC cells the effect of hydrogen-enriched medium against DON induced cell injury. The manuscript is in line with a series op papers of the same authors. Moreover, the results are in line with a considerable number of earlier manuscripts, which indicated already that hydrogen application (in vivo) or incubations of cells in hydrogen-enriched cell culture medium exert a strong antioxidant effect. The current results are in line with these previous reports.
A shortcoming of the current manuscript is the lack of any explanation of the nature (and procedure) of H2-enrichment of the cell culture medium. Hence the reader needs to study other references to get some information on these procedures. Assuming that H2 saturated medium was used, it would have been more elegant to test different concentrations of DON to demonstrate a dose/concentration dependent effect (a standard requirement in all interention studies).
In summary, the authors are urged to
Add the procedure by which the H2-enriched cell culture medium was produced to the M&M section. Moreover, a measurement indicating the actual concentration of H2 in the medium should be added (preferentially at the beginning and end of the experimental period). Present the rationale to use a single DON concentration and motivate the selection of 5 microM of DON as representative mycotoxin concentration. In the discussion, several literature citations present DON-concentrations in ng/ml. Hence a re-calculation of the experimental concentrations should be added as well to the M&M section to improve readability.The introduction would benefit from an extension and more detailed description (with additional references) of the anti-oxidant properties of H2, which have been studied intensively before and in similar in vitro approaches (albeit with other cell types).
Editorial comments:
Please add to the legends of all figures the concentration of DON and the concentrat6ion of H2, or use the term H2-saturated medium where appropriate.
Figure 1: The presentation of photographs from cells is unusual (in such a low magnification) and could be deleted, as the relevant results are expressed with the outcome of the MTT assay and the LDH measurements.
Author Response
Dear Reviewer, we are very grateful for your kindly comments and valuable advice. Please see the attachment about the response letter, thank you very much.

Reviewer 3 Report
Referee evaluation of the manuscript “Protective Role of Hydrogen Gas on Oxidative Damage and Apoptosis in Intestinal Porcine Epithelial Cells (IPEC-J2) Induced by Deoxynivalenol- A Preliminary Study” (Ref. no.: TOXINS-650610) Summarized: The manuscript is generally well conceived. Its results are correct. Indeed the biochemical measurements, the way how they were carried olut, and the results and there interpretation is acceptable. My basic concern is double: - I clearly understand that this is an in vitro setting. But in theory the results must or might be interpolated to the in vivo systems as well. How is it viable to use H2 gas to reduce the toxic and oxidative effects of a mycotoxin in vivo? Indeed the anti-oxidative effect of a non-oxygen gas, that is ab ovo totally reductive ... is somewhat fully axiomatic. - The hydrogen enriched medium is a non-physiological medium itself. I believe all results of the authors, but I have to criticize the basic hypothesis. The presence of a well-known gas that is not oxidative is axiomatically reducing oxidative effects. I believe that all inert gases have slight but similar effects, like N, He, Ar, etc. I do not see that this preliminary study may or might have any effective or novel continuation. In summary, due to the lack of novelty of the hypothesis, to the instability of the basic conception, to the axiomatic approach, and ultimately due to the full lack of interpolation possibility to in vivo systems I do not recommend the acceptance of this manuscript for publication in TOXINS.
Author Response

(The authors gave the same response as above.)

Round 2
Reviewer 3 Report
Please see the attached file

Author Response
Dear Reviewer, we are very appreciated for your kindly comments and valuable suggestions. Please see the attachment about the response letter, thank you very much.
